# A 12-Week, Single-Centre, Randomised, Double-Blind, Placebo-Controlled, Parallel-Design Clinical Trial for the Evaluation of the Efficacy and Safety of *Lactiplantibacillus plantarum* SKO-001 in Reducing Body Fat

**DOI:** 10.3390/nu16081137

**Published:** 2024-04-11

**Authors:** Seon Mi Shin, Jeong-Su Park, Sang Back Kim, Young Hee Cho, Hee Seo, Hak Sung Lee

**Affiliations:** 1Department of Internal Medicine, College of Korean Medicine, Semyung University, Semyeong-ro 65, Jecheon-si 27136, Republic of Korea; 2Department of Preventive Medicine, College of Korean Medicine, Semyung University, Semyeong-ro 65, Jecheon-si 27136, Republic of Korea; suyahpark@gmail.com; 3Food Science R&D Center, Kolmar BNH Co., Ltd., 61, Heolleung-ro 8-gil, Seocho-gu, Seoul 06800, Republic of Korea; m302@kolmarbnh.co.kr (S.B.K.); cho6452@kolmarbnh.co.kr (Y.H.C.); seohee@kolmarbnh.co.kr (H.S.); mildpeople@kolmarbnh.co.kr (H.S.L.)

**Keywords:** *Lactiplantibacillus plantarum* SKO-001, placebo-controlled study, randomised controlled trial, obesity, nutritional supplement

## Abstract

There is growing evidence linking gut microbiota to overall health, including obesity risk and associated diseases. *Lactiplantibacillus plantarum* SKO-001, a probiotic strain isolated from *Angelica gigas*, has been reported to reduce obesity by controlling the gut microbiome. In this double-blind, randomised clinical trial, we aimed to evaluate the efficacy and safety of SKO-001 in reducing body fat. We included 100 participants randomised into SKO-001 or placebo groups (1:1) for 12 weeks. Dual-energy X-ray absorptiometry was used to objectively evaluate body fat reduction. Body fat percentage (*p* = 0.016), body fat mass (*p* = 0.02), low-density lipoprotein-cholesterol levels (*p* = 0.025), and adiponectin levels (*p* = 0.023) were lower in the SKO-001 group than in the placebo group after 12 weeks of SKO-001 consumption. In the SKO-001 group, the subcutaneous fat area (*p* = 0.003), total cholesterol levels (*p* = 0.003), and leptin levels (*p* = 0.014) significantly decreased after 12 weeks of SKO-001 consumption compared with baseline values. Additionally, SKO-001 did not cause any severe adverse reactions. In conclusion, SKO-001 is safe and effective for reducing body fat and has the potential for further clinical testing in humans.

## 1. Introduction

Obesity is a major health issue that is strongly associated with metabolic diseases [1]. Recent research has suggested that gut microbiota can influence overall health, including obesity risk [2]. There is growing interest in using natural substances that can help reduce body fat or obesity [3]. Nutrient absorption in the gut is correlated with metabolic diseases, such as obesity, insulin resistance, and diabetes, and improvements in obesity-related comorbidities, such as weight loss following gastric surgery, have been attributed to surgery-induced changes in the microbiome and metabolic changes in the gut microbiota [4,5,6]. In particular, probiotics, i.e., beneficial living microorganisms that help maintain intestinal health, are recognised for their anti-obesity effects. Most probiotics are lactic acid bacteria that are non-toxic and non-pathogenic. They produce organic acids such as acetic acid in the intestines and acidify the intestinal environment, thereby reducing harmful bacteria. When consumed as a functional food ingredient, probiotics are known to promote the growth of lactic acid bacteria, inhibit harmful bacteria, and facilitate healthy bowel movements [7]. They are also beneficial for vaginal health, immune regulation, intestinal health, skin protection against ultraviolet damage, and the maintenance of skin moisture [8]. Mechanistic studies have indicated that the gut microbiome can regulate energy balance, fat storage, neurohormonal functions, and the immune system [9,10]. Thus, manipulating the composition of the gut microbial ecosystem has been proposed as a novel approach to facilitate weight loss and prevent obesity. One of the emerging potentially effective treatments for obesity prevention and/or treatment is the intake of probiotics to alter the composition of the gut microbiome [11]. Previous studies have reported that the probiotic *Lactiplantibacillus plantarum*, SKO-001, K50, KY1032, and LMT1-48 can reduce obesity and improve weight- or body fat-related indicators, possibly via the regulation of lipid metabolism [12,13,14,15,16].

*L. plantarum* is a gram-positive bacillus that produces lactic acid by fermenting arabinose, glucose, fructose, galactose, maltose, sucrose, and dextran. It is generally recognised as safe and exhibits strong probiotic properties, with several strains commercialised as health-promoting supplements and functional foods [13]. *L. plantarum* SKO-001 (SKO-001) is an *L. plantarum* strain isolated from the root surface of *Angelica gigas* Nakai, a perennial plant used as a medicinal herb in China and Korea, and is deposited in the Biological Resources Centre of the Korea Research Institute of Bioscience and Biotechnology (Accession No. KCTC 14816BP). The full-length genome sequence has been analysed and deposited in the U.S. National Center for Biological Information. Genome sequence registration has also been completed [13].

In vitro studies have demonstrated that SKO-001 treatment of differentiated adipose progenitor cells (3T3-L1) reduced fat accumulation, decreased fat differentiation and synthesis-related proteins, and inhibited the activity of pancreatic lipase [13]. In addition, in an in vivo study using an animal model of obesity induced by a high-fat diet, SKO-001 treatment reduced body fat, decreased body weight, decreased factors related to fat synthesis and differentiation, increased factors regulating energy metabolism, increased thermogenic factors, and reduced liver fat accumulation [13].

The purpose of this study was to determine the safety of *L. plantarum* and its influence on the gut environment to affect weight loss or body fat loss and to provide clinical evidence that the gut microbiota influence metabolism in humans.

## 2. Materials and Methods

### 2.1. Study Design and Participants

This protocol (version 1.2, publication date 1 January 2023) was approved by the Institutional Review Board of Seymour University Oriental Medicine Hospital (approval number SMJOH-2022-08) and registered with the Korea Clinical Research Information Center (KCT0008871) to test the effectiveness of dietary supplements in reducing body fat.

This randomised, double-blind, placebo-controlled clinical trial recruited healthy participants through advertisements on the hospital website and bulletin boards at Seymyeong University Oriental Medicine Hospital (Jecheon, Chungcheongbuk-do, South Korea) until the target sample size was achieved. Individuals willing to participate visited the internal medicine department and were screened according to the participant selection criteria (Section 2.2). The first participant was enrolled in September 2022. The duration of study participation was 14 weeks, including a washout period of up to 2 weeks and a safety assessment conducted 2 weeks after the last visit. A washout period of up to 2 weeks was required if the participant had a history of concomitant use of prohibited medications or foods. Participants were randomised to either a treatment (SKO-001) or placebo group at the second visit (within 2 weeks of the first visit), which was considered the baseline time point.

Prior to randomisation, the inclusion and exclusion criteria were rechecked, and participants who met the criteria were enrolled. A baseline assessment was performed, and participants were provided with a 33-day supply of the intervention or placebo. Follow-up visits occurred 28 (visit 2), 56 (visit 3), and 84 (visit 4) days after baseline assessment (visit 5) (Figure 1). In addition, a 5-day visit gap was allowed. During visits 3–5, vital signs, medical history/concurrent medication screening, and efficacy and safety were assessed. On visits 1 and 5, laboratory and pregnancy tests were performed. The investigator informed the participants of their clinical visit schedule.

### 2.2. Participant Selection Criteria

The inclusion criteria were as follows: (1) age 19–60 years, (2) body mass index (BMI) ≥25 kg/m^2^ and <30 kg/m^2^, and (3) the ability to provide written informed consent. The exclusion criteria were as follows: (1) a history of drug hypersensitivity to medicines, food ingredients, herbal extracts, dietary supplements, or lactic acid bacteria; (2) severe cardiovascular, immune, respiratory, hepatobiliary, renal, urinary, nervous, or muscular conditions or currently undergoing treatment for conditions such as skeletal, psychiatric, infectious diseases, and malignant tumours; (3) taking prebiotics and probiotics similar to the interventional formulation within 4 weeks of screening; (4) participation in a commercial obesity treatment programme and weight loss within 3 months of screening; (5) taking drugs that affect body weight within 3 months of screening (fat absorption inhibitors, appetite suppressant, beta-blocker, glucagon-like peptide 1 (GLP-1) receptor agonist, health functional food related to obesity improvement/supplements, or psychiatric drugs such as antidepressants, diuretics, contraceptives, steroids, and female hormonal drugs); (6) weight change of >10% within 3 months of screening; (7) undergone gastroplasty, omentectomy, or other weight loss surgeries within 6 months of screening; (8) obese or overweight due to endocrine diseases such as hypothyroidism and Cushing’s syndrome; (9) thyroid-stimulating hormone (TSH) < 0.1 μU/mL and > 10 μU/mL; (10) aspartate aminotransferase (AST) or alanine transaminase (ALT) at least three times higher than the upper limit of normal range; (11) creatinine levels more than twice the upper normal limit in the testing institute; (12) uncontrolled hypertension (blood pressure ≥ 160/100 mmHg, or changing or starting new medication for hypertension within 3 months); (13) fasting blood glucose ≥ 126 mg/dL, diagnosed diabetes mellitus, or taking oral hypoglycaemic agents or insulin; (14) severe gastrointestinal symptoms such as heartburn and indigestion; (15) serious psychiatric condition (e.g., schizophrenia, epilepsy, anorexia, or bulimia); (16) musculoskeletal conditions that rendered the person unable to exercise; (17) women who were pregnant or lactating, or planning to become pregnant during the study period; (18) a history of participation in another trial or ingestion of an investigational drug within 3 months of screening; and (19) deemed by the principal investigator (or delegated investigator) to be inappropriate to participate for any other reason, e.g., those considered unfit for exercise. We did not enrol those who took antibiotics within 4 weeks of starting the study, and those who took antibiotics after 4 weeks were enrolled in the study. We also excluded participants if they were taking antibiotics for >3 days or had a gastrointestinal condition that could affect the absorption of food or medication.

The criteria for withdrawal from clinical trials were as follows: (1) serious adverse events occurred during the clinical trial; (2) taking medications or diet foods that may affect body fat and weight other than the study food during the course of the human clinical trial (lemon verbena extract complex, soursop extract, moringa orange extract powder, grapefruit extract complex, cissus extract, fermented vinegar pomegranate complex, lactoferrin, diglyceride-containing oil, green tea extract, garcinia cambogia extract, etc.); (3) evaluating results becoming challenging if participants deviated from the prescribed method or schedule of consuming the test food; (4) participants withdrawing consent to participate in the clinical trial; (5) participants having difficulty following up due to personal circumstances during the study; (6) When, in the opinion of the investigator, the participant is no longer able to participate in the study; and (7) confirmation of pregnancy.

### 2.3. Contraindicated Drugs/Foods

The following medications/nutraceuticals were prohibited because they might directly or indirectly affect the safety of participants or the results of this study, and subjects taking the following medications were excluded from the study.

(1) Contraindicated drugs: fat absorption inhibitors, appetite suppressants, beta-blockers, GLP-1 receptor agonists, psychiatric drugs such as antidepressants, diuretics, contraceptives, steroids, female sex hormones, etc. (2) Dietary supplements: lemon verbena extract and other complexes, sarsaparilla extract, moro orange extract powder, green apple extract apple phenone, fermented yulipi extract powder, seaweed extract, hydrangea leaf hydrothermal water extract, grapefruit extract and other complexes, wood thorn extract, Lactobacillus complex HY760 + KY1032, sesame leaf extract, cissus extract, green coffee bean extract, *Lactobacillus gasseri* BNR17, finger root extract powder, wild mango seed extract, fermented vinegar pomegranate complex, Boicha extract, medium chain fatty acids, lactoferrin, green mate extract, wakame seaweed complex extract, poria cocos leaf extract powder, hornbeam taffeta peptide, L-carnitine tartrate, lemon balm extract mixed powder, coleus forskohlii extract, diglycerides, soybean embryo hydrothermal extract complex, hibiscus complex extract, green tea extract, conjugated linoleic acid, garcinia cambogia extract, and chitosan/chitooligosaccharide. (3) Other drugs or health-functional foods that may affect the absorption, metabolism, or excretion of the test food according to the judgement of the principal investigator.

### 2.4. Intervention

Based on previous in vitro and in vivo studies [7], the daily intake of SKO-001 was set to 2 × 10^10^ CFU/day (Appendix A)**.** Both interventional and placebo formulations were consumed orally once daily, 1 capsule, for 12 weeks. In the placebo group, a capsule with the same shape and texture was provided to the participants (Figure 2).

### 2.5. Randomisation and Blinding

Permuted block randomisation was used, and a statistician sequentially applied the permutation of random numbers generated in the R software (v4.2.0; R Project for Statistical Computing, Vienna, Austria) starting with test participant number 1. A random allocation table was prepared separately so that the SKO-001 and placebo groups could be assigned treatment in a 1:1 ratio (for instance, No. 1 = A = SKO-001 group, No. 2 = B = placebo group). The block size was set to 2 or 4. The participant number was a total of three digits starting with R and had a certain rule (XX Hospital: R001). The number assigned to each test participant served as the participant identification code until the clinical trial was concluded.

To ensure double blinding, participants were provided with clinical trial intervention labels as mentioned in the protocol during the study. The formulation produced as a placebo was identical in appearance and characteristics to SKO-001, with no visible differences or significant weight differences. The same labelling also ensured that the double blinding of participants and investigators was maintained. The investigator or controlling pharmacist provided the interventional formulation to the participants in random order. The sponsor maintained the allocation of randomisation codes to the arms. The randomisation codes were not disclosed until the end of the study unless a serious adverse event necessitated access to the codes.

### 2.6. Endpoints

The primary endpoint was the alteration in body fat percentage, evaluated using dual-energy X-ray absorptiometry (DEXA; GE Healthcare, Madison, WI, USA) after 12 weeks (visit 5) compared with baseline (visit 2). The secondary endpoints were as follows: (1) changes in lean mass and body fat mass measured using DEXA at the 12-week mark; (2) alterations in total fat area, subcutaneous fat area, visceral fat area, and the ratio of visceral fat area to subcutaneous fat area assessed using abdominal computed tomography (CT; Siemens Healthineers, Erlangen, Germany) at 12 weeks compared to baseline; (3) shifts in body weight, BMI, waist circumference, hip circumference, and waist/hip circumference ratio at 4, 8, and 12 weeks from the start; and (4) variations in blood lipid levels (total cholesterol, LDL-cholesterol [C], triglycerides [TG], and HDL-C), as well as levels of leptin, adiponectin, and insulin at the 12-week mark relative to baseline. DEXA was used as the primary tool to gauge body fat percentage at both baseline and week 12. Total fat mass was evaluated using the LUNAR Prodigy Vision scanner (software version 6.70; General Electric Medical Systems, Madison, WI, USA) in the supine position with a whole-body DEXA scanner. Specific quantification of body fat mass involved standard soft tissue measurement techniques, with the fat regions in the chest, abdomen, and pelvis delineated using a whole-body DEXA scanner. Detailed body composition analyses were performed via DEXA, primarily focusing on the trunk region, which is pertinent to abdominal obesity. Additionally, abdominal CT was performed to measure visceral, subcutaneous, and total abdominal fat areas and the ratio of visceral to subcutaneous fat at baseline and week 12, with CT conducted between the fourth and fifth lumbar vertebrae. The assessment of physical activity and dietary habits included the use of the Korean version of the short-form International Physical Activity Questionnaire at visits 2 and 5, along with dietary analysis via the 24-h recall method. Dietary patterns were further explored through total calorie (kcal) analysis using the Samsung Health application, focusing on participants consuming >1500 kcal and those with physical activity levels below 3000 METs/week.

### 2.7. Safety

The safety endpoints were adverse events, vital signs, and clinicopathological examinations. If an adverse event occurred in a patient, then the severity, outcome, relevance, and related measures were recorded. Vital signs (blood pressure, pulse rate, and temperature) were recorded to assess patient safety. During vital sign assessments, participants rested for 5 min, and their vital signs were measured while they remained in the same position. Clinicopathological tests included haematology, blood biochemistry, and urinalysis. For these tests, the participants were required to fast for 8 h on the day of blood collection.

### 2.8. Sample Size Calculation

The number of participants was calculated using the results reported by Cho et al. [17], a clinical trial that showed a significant effect on body fat percentage (body fat) measured using DEXA, an equivalent of this human clinical trial among existing studies. The standard deviation of the difference is not presented; thus, it was estimated using a significance probability (*p*-value) of 0.018 at week 12. The estimated pooled standard deviation of the mean difference was approximately 2.14. Using this calculated change in body fat mass, the required number of participants to achieve a significance level of 5% and a power of 80% was determined to be 25 per group. Accounting for an anticipated dropout rate of 30%, we planned to enrol 50 participants per group (resulting in a total of 100 participants).

### 2.9. Statistical Analyses

Descriptive statistics (mean, standard deviation, median, minimum, and maximum) are presented for the efficacy evaluation index at baseline at 12 weeks and its changes. Comparisons between groups were analysed using independent-sample *t*-tests if the data were normally distributed and the Wilcoxon rank sum test if normality was not satisfied. However, only when there was a difference between groups at baseline, analysis of covariance (ANCOVA) was performed with the evaluation item as the baseline covariate. If there was a significant difference (*p* < 0.05 using *t*-test or Wilcoxon rank sum test) in the amount of change in the SKO-001 group compared with the placebo group, the interventional formulation was evaluated as having a valid effect. Within-group comparisons were analysed using paired *t*-tests if the data were normally distributed, and Wilcoxon signed-rank test if not.

All adverse events were standardised to the “System Organ Class” and “Preferred Term” using MedDRA Version 25.0. In addition, the Common Terminology Criteria for Adverse Events (CTCAE, Version 5.0) were used to compare the severity of the adverse event with the relevance of the study drug, and the number of patients who experienced at least one adverse event, adverse drug reaction, or serious adverse event during the study was presented as the number of patients (incidence rate) and the number of cases. In addition, 95% confidence intervals of expression rates are presented, and the statistical significance of the expression rates between treatment groups was tested using the chi-square test or Fisher’s exact test.

For continuous data from clinicopathologic examinations and on vital signs, descriptive statistics (mean, standard deviation) are presented for mean changes from baseline, and the data were tested for normality and compared between groups using independent-samples *t*-tests (if normal) or Wilcoxon rank sum test (if non-normal). For within-group comparisons, data were tested for normality and analysed using paired *t*-tests if normality was met and Wilcoxon signed-rank test if normality was not met. Descriptive statistics (frequency and percentage) are presented for categorical data, and comparisons between groups were performed using the chi-square test or Fisher’s exact test. All statistical analyses were performed using SAS (version 9.4; SAS Institute, Cary, NC, USA).

## 3. Results

### 3.1. Participant Characteristics

The first participant was screened on 14 November 2022, and the last participant on 14 June 2023. In total, a hundred and seven participants were evaluated, of whom seven were excluded. Four participants were excluded from screening because they did not meet the inclusion criteria (one with ALT or AST greater than three times the upper limit of normal; two with fasting glucose greater than 126 mg/dL, TSH less than or equal to 0.1 μU/mL, and TSH greater than or equal to 10 μU/mL), and three were excluded because they withdrew their consent before randomisation. Of the 100 participants finally included, 50 were randomly assigned to the SKO-001 and placebo groups each, and the clinical trial was conducted for 12 weeks. Two participants from the SKO-001 group and one from the placebo group dropped out, and a total of ninety-seven participants finally completed the clinical trial (Figure 3, Table 1).

### 3.2. Study Endpoints

The body fat percentage (primary endpoint) after 12 weeks of SKO-001 consumption decreased in the SKO-001 group (change value = −0.19 ± 1.36%) but increased in the placebo group (0.48 ± 1.23%), with the difference between the groups being statistically significant (*p* = 0.016). The pre- and post-intervention comparison showed a significant increase in the change from baseline to 12 weeks of intervention in the placebo group (*p* = 0.011) (Appendix A, Figure 4).

Based on the outcomes of the detailed DEXA variables after 12 weeks of intervention, an inter-group analysis of the change in body fat percentage of the trunk and android and body fat mass of the trunk was performed. Significant differences were found in the trunk and android body fat percentage (*p* = 0.005 and *p* = 0.010, respectively) and trunk body fat mass (*p* = 0.017). The trunk body fat percentage decreased in the SKO-001 group (change value = −0.23 ± 1.80%) but increased in the placebo group (change value = 0.83 ± 1.78%). Android fat percentage decreased in the SKO-001 group (change value = −0.33 ± 2.35%) but increased in the placebo group (change value = 0.91 ± 2.14%); trunk body fat mass decreased in the SKO-001 group (change value = −118.22 ± 805.98 g) but increased in the placebo group (change value = 300.79 ± 851.86 g). The outcomes of the inter-group analysis by time point (visits 2 and 5) were statistically significant (Appendix A, Figure 4). However, a within-group analysis did not show significant differences. A significant increase was observed in the placebo group (trunk body fat percentage—SKO-001 group: *p* = 0.388, placebo group: *p* = 0.002; android body fat percentage—SKO-001 group: *p* = 0.356, placebo group: *p* = 0.005; trunk body fat mass—SKO-001 group: *p* = 0.331, placebo group: *p* = 0.019) (Appendix A, Figure 4).

The secondary endpoints were changes in body fat, LDL-C, and adiponectin levels. The change in body fat levels decreased in the SKO-001 group (change value = −169.51 ± 1168.26 g) but increased in the placebo group (change value = 350.38 ± 1216.37 g), and the between-group analysis confirmed a significant between-group difference (*p* = 0.020) (Appendix A). The LDL-C level decreased in the SKO-001 group (change value = −6.31 ± 19.91 mg/dL) but increased in the placebo group (change value = 3.23 ± 20.38 mg/dL), and group analysis revealed a significant difference between the groups (*p* = 0.025). The level of adiponectin increased in the SKO-001 group (change value = 454.36 ± 2270.17 ng/mL) but decreased in the placebo group (change value = −628.20 ± 2133.35 ng/mL), and between-group analysis showed a significant difference in this regard (*p* = 0.023) (Table 2). Thus, after 12 weeks of SKO-001 consumption, body fat mass and LDL-C levels significantly decreased in the SKO-001 group compared with those in the placebo group, and adiponectin levels significantly increased (Table 2).

Inter-group analysis of the visit days and the amount of change in leptin levels showed no significant difference. Within-group analysis showed a significant decrease in the SKO-001 group after 12 weeks of SKO-001 consumption, with a change of −4.03 ± 10.53 ng/mL; however, no significant difference was found in the placebo group (SKO-001 group: *p* = 0.014, placebo group: *p* = 0.093) (Table 2). The change in body fat mass decreased in the SKO-001 group (change value = −169.51 ± 1168.26 g) but increased in the placebo group (change value = 350.38 ± 1216.37 g). Inter-group analysis confirmed a significant difference between the groups in this regard (*p* = 0.020) (Table 3).

Inter-group analysis of the amount of change did not reveal significant between-group differences in lean mass, visceral fat area, body weight, BMI, waist circumference, hip circumference, waist/hip circumference ratio, TG, HDL-C, and insulin values. Moreover, there was no significant difference in within-group analysis (Table 2, Table 3 and Appendix A).

Regarding the total abdominal fat area, within-group analysis showed no significant difference in the SKO-001 group, but a decreasing trend was observed. Moreover, a significant decrease was observed in the placebo group (SKO-001 group: *p* = 0.059, placebo group: *p* = 0.028) (Table 3). Inter-group analysis showed no significant difference in the timing and amount of change in the visceral fat/subcutaneous fat area ratio. Within-group analysis showed no significant difference in the SKO-001 group; however, in the placebo group, there was a significant increase of 0.07 ± 0.15 cm^2^ after 12 weeks of treatment (SKO-001 group: *p* = 0.132, placebo group: *p* = 0.002) (Table 3). Regarding the timing and amount of change in the subcutaneous fat area, inter-group analysis showed no significant difference. Within-group analysis also showed a significant decrease in both groups (SKO-001 group, *p* = 0.008; placebo group, *p* = 0.002) (Table 3). For inter-group analysis of the timing and amount of change in the total cholesterol level, no significant difference was found. Within-group analysis showed a significant decrease in the SKO-001 group, with no significant difference in the placebo group (SKO-001 group: *p* = 0.003, placebo group: *p* = 0.194) (Table 2). 

Calorie intake was compared between the SKO-001 and placebo groups (Appendix A). Between-group analysis did not show significant differences at baseline or 12 weeks after consuming the investigational food, nor did the amount of change. Within-group analysis also showed no significant differences.

The results of the comparison of efficacy evaluation variables between the SKO-001 and placebo groups among participants whose calorie intake exceeded 1500 kcal at visit 2 were as follows: Inter-group analysis of changes in efficacy evaluation variables after 12 weeks of SKO-001 consumption showed significant differences in body fat percentage (*p* = 0.030), body fat mass (*p* = 0.004), LDL-C (*p* = 0.012), and adiponectin (*p* = 0.004). Moreover, body fat percentage decreased in the SKO-001 group (change value = −0.26 ± 1.50%) but increased in the placebo group (change value = 0.57 ± 1.20%); body fat mass decreased in the SKO-001 group (change value = −306.89 ± 1239.03 g) but increased in the placebo group (change value = 521.26 ± 1141.93 g); LDL-C levels decreased in the SKO-001 group (change value = −9.70 ± 19.63 mg/dL) but increased in the placebo group (change value = 5.00 ± 21.66 mg/dL); and adiponectin levels increased in the SKO-001 group (change value = 926.17 ± 1997.79 ng/mL) but decreased in the placebo group (change value = −824.22 ± 2019.01 ng/mL). The results of the inter-group analysis by time point were not statistically significant.

In the within-group analysis before and after SKO-001 consumption, there was no significant difference in body fat percentage and body fat mass in the SKO-001 group, but a significant increase was found in the placebo group (body fat percentage—SKO-001 group: *p* = 0.385, placebo group: *p* = 0.021; body fat mass—SKO-001 group: *p* = 0.645, placebo group: *p* = 0.025). There was a significant decrease in the LDL-C level in the SKO-001 group, but no significant difference was observed in the placebo group (SKO-001 group: *p* = 0.016, placebo group: *p* = 0.241). Adiponectin levels were not significantly different in the SKO-001 group, but a significant decrease was observed in the placebo group (SKO-001 group: *p* = 0.111, placebo group: *p* = 0.044). Regarding the other variables, no significant differences were found in the inter-group analysis of the amount of change between the SKO-001 and placebo groups. The results of inter-group analysis of the change in efficacy evaluation variables among participants with a calorie intake of ≤1500 kcal (as of visit 2) showed no significant variables (Appendix A).

The results of the comparison of efficacy evaluation variables between the SKO-001 and placebo groups among participants whose average calorie intake exceeded 1500 kcal at baseline (visit 2) and 12 weeks (visit 5) are as follows: Inter-group analysis of changes in efficacy evaluation variables after 12 weeks of SKO-001 consumption showed significant differences in body fat percentage (*p* = 0.017), body fat mass (*p* = 0.007), weight (*p* = 0.023), BMI (*p* = 0.035), LDL-C (*p* = 0.038), and adiponectin (*p* = 0.005). Body fat percentage decreased in the SKO-001 group (change value = −0.34 ± 1.37%) but increased in the placebo group (change value = 0.58 ± 1.24%); body fat mass also decreased in the SKO-001 group (change value = −474.88 ± 1291.21 g) but increased in the placebo group (change value = 531.54 ± 1190.79 g); body weight decreased in the SKO-001 group (change value = −0.61 ± 1.36 kg) but increased in the placebo group (change value = 0.35 ± 1.50 kg); BMI decreased in the SKO-001 group (change value = −0.22 ± 0.51 kg/m^2^) but increased in the placebo group (change value = 0.11 ± 0.56 kg/m^2^); the LDL-C level decreased in the SKO-001 group (change value = −7.20 ± 19.25 mg/dL) but increased in the placebo group (change value = 5.50 ± 22.21 mg/dL); the adiponectin level increased in the SKO-001 group change value = (757.14 ± 2154.29 ng/dL) but decreased in the placebo group (change value = -831.58 ± 1522.96 ng/dL). The results of inter-group analysis by time point were not statistically significant.

The intra-group analysis before and after treatment showed no significant difference in body fat percentage and body fat mass in the SKO-001 group, but a significant increase was found in the placebo group (body fat percentage—SKO-001 group: *p* = 0.227, placebo group: *p* = 0.030, body fat mass—SKO-001 group: *p* = 0.078, placebo group: *p* = 0.039). A significant decrease in body weight was observed in the SKO-001 group, but no significant difference was observed in the placebo group (SKO-001 group, *p* = 0.034; placebo group, *p* = 0.266). There were no significant differences in BMI and LDL-C levels between the two groups. There was no significant difference in adiponectin levels in the SKO-001 group; however, a significant decrease was observed in the placebo group (SKO-001 group, *p* = 0.092; placebo group, *p* = 0.014). Regarding the other variables, no significant differences were found in the inter-group analysis of the extent of change between the SKO-001 and placebo groups (Appendix A).

According to the inter-group analysis of the amount of change in efficacy evaluation variables among participants with a caloric intake of 1500 kcal or less (based on the average of visits 2 and 5), the change in visceral fat/subcutaneous fat area ratio was −0.01 ± 0.11 in the SKO-001 group and 0.09 ± 0.17 in the placebo group, with a significant difference (*p* = 0.030). However, the within-group analysis showed no significant change in the SKO group but a significant increase in the placebo group (SKO-001 group, *p* = 0.754; placebo group, *p* = 0.019).

In the comparison of efficacy evaluation variables among participants with physical activity <3000 METs/week at visit 2, inter-group analysis after 12 weeks of SKO-001 consumption showed significant between-group differences in body fat percentage (*p* = 0.027), body fat mass (*p* = 0.019), TG (*p* = 0.043), and adiponectin (*p* = 0.048). Body fat percentage decreased in the SKO group (change value = −0.18 ± 1.42%) but increased in the placebo group (change value = 0.54 ± 1.22%); body fat mass decreased in the SKO-001 group (change value = −222.76 ± 1211.01 g) but increased in the placebo group (change value = 392.39 ± 1241.87 g); TG increased in the SKO-001 group (change value = 13.76 ± 48.36 mg/dL) but decreased in the placebo group (change value = −14.45 ± 49.50 mg/dL); and adiponectin increased in the SKO-001 group (change value = 390.03 ± 2430.84 ng/mL) but decreased in the placebo group (change value = −652.65 ± 2224.03 ng/mL). The results of inter-group analysis by time point were not statistically significant.

In the within-group analysis before and after treatment, there was no significant difference in body fat percentage in the SKO-001 group; however, a significant increase was observed in the placebo group (SKO-001 group, *p* = 0.450; placebo group, *p* = 0.017). There were no significant differences in body fat mass, TG, and adiponectin levels between the two groups. Regarding the other variables, no significant differences were found in the inter-group analysis of the extent of change between the SKO-001 and placebo groups (Appendix A).

The inter-group analysis of the change in efficacy evaluation variables among participants with physical activity of ≥3000 METs/w (as of visit 2) indicated that the change in the HDL-C level was statistically significant (*p* = 0.047); its level was −0.62 ± 6.46 mg/dL in the SKO-001 group; the placebo group showed a change of 2.21 ± 4.59 mg/dL, indicating that the level in the SKO-001 group decreased more than that in the placebo group. Within-group analysis revealed no significant differences between the groups.

In the comparison of efficacy evaluation variables between the groups among participants with physical activity of <3000 METs/w based on the average at baseline (visit 2) and 12 weeks (visit 5), the inter-group analysis of the change in efficacy evaluation variables after 12 weeks of SKO-001 consumption indicated significant differences in body fat percentage (*p* = 0.027), body fat mass (*p* = 0.019), TG (*p* = 0.043), and adiponectin (*p* = 0.048). Body fat percentage decreased in the SKO-001 group (change value = −0.18 ± 1.41%) but increased in the placebo group (change value = 0.56 ± 1.32%); body fat mass decreased in the SKO group (change value = −172.24 ± 1225.58 g) but increased in the placebo group (change value = 339.97 ± 1308.69 g); and the TG level increased in the SKO-001 group (change value = 11.89 ± 49.09 mg/dL) but decreased in the placebo group (change value = −19.26 ± 46.48 mg/dL). The SKO-001 group had lower body fat percentage and body fat mass and higher TG levels, whereas the placebo group had significantly increased body fat percentage and decreased TG levels. The results of the inter-group analysis by time point were not statistically significant.

Within-group analysis showed no significant difference in body fat percentage in the SKO-001 group, but a significant increase was observed in the placebo group (SKO-001 group, *p* = 0.450; placebo group, *p* = 0.017). There was no significant difference in body fat mass between both groups, and TG levels showed no significant difference in the SKO-001 group but showed a significant decrease in the placebo group (SKO-001 group: *p* = 0.261, placebo group: *p* = 0.005). Regarding other variables, no significant differences were found in the inter-group analysis of the extent of change between the groups.

The inter-group analysis of the change in efficacy evaluation variables among participants with physical activity of >3000 METs/w (based on the average of visits 2 and 5) indicated that the change in the adiponectin level was statistically significant (*p* = 0.031); its level in the SKO-001 group increased more than that in the placebo group (SKO-001 group, 1588.62 ± 2139.09 ng/mL; placebo group, −523.25 ± 1350.52 ng/mL). Within-group analysis showed no significant differences between both groups in this regard (Appendix A).

### 3.3. Safety

Adverse reactions occurred in 27 participants (46 cases) in the SKO-001 group and 28 (46 cases) in the placebo group. There was one serious adverse event in the placebo group; the participant was hospitalised for back pain and subsequently dropped out of the trial. The severity of adverse reactions was mild in ninety cases and moderate in one case (Appendix A). Regarding adverse reactions, eighty-five cases were completely cured and seven were in progress. Action was taken for 50 participants who experienced adverse reactions. A combination of therapeutic drugs was administered to seventy-five cases, and non-drug treatment was administered to one. However, this was not related to the intervention. The inter-group analysis of adverse reaction severity, outcome, causality, and measures related to the intervention did not yield significant results (Table 4).

Clinical pathological tests included haematological, blood biochemical, and urine tests. As a result of the inter-group analysis, significant differences were found in changes in the haematological test findings: the haemoglobin level at visit 2 (*p* = 0.016), platelet count at visit 5 (*p* = 0.028), neutrophil count at visit 5 (*p* = 0.021), neutrophil change (*p* = 0.028), lymphocyte count at visit 5 (*p* = 0.049), and lymphocyte change (*p* = 0.027). There were no differences in blood biochemical and urine test findings between the groups (Appendix A). The within-group analysis of haematological test findings showed a significant decrease in the counts of white blood cells (*p* = 0.040) and red blood cells (*p* = 0.000) in the SKO-001 group, haemoglobin levels in the SKO-001 (*p* = 0.006) and placebo (*p* = 0.048) groups, and haematocrit levels in the SKO-001 (*p* = 0.001) and placebo (*p* = 0.027) groups. In the blood biochemical tests, there were significant decreases in AST levels in the SKO-001 (*p* = 0.006) and placebo (*p* = 0.018) groups, ALT levels in the SKO-001 group (*p* = 0.013), gamma-glutamyltransferase levels in the SKO-001 group (*p* = 0.046), alkaline phosphatase levels in the SKO-001 (*p* = 0.001) and placebo (*p* = 0.009) groups, and C-peptide levels in the placebo group (*p* = 0.006), as well as a significant increase in total bilirubin in the placebo group (*p* = 0.012) (Appendix A). The significant differences in the findings of inter-group and within-group analyses of urine tests were within the normal range, indicating that they were not clinically significant (Appendix A and Appendix A). There was no significant difference in vital signs in both inter-group and within-group analyses (Appendix A).

### 3.4. Physical Activity and Diet

There were no significant between-group differences in dietary intake before (0 weeks) and after (12 weeks) SKO-001 or placebo consumption.

## 4. Discussion

The results of the present study showed a significant difference in changes in body fat percentage between the SKO-001 and placebo groups. In particular, the placebo group showed an increase in body fat percentage, whereas the SKO-001 group showed a decrease in body fat percentage. Changes in body fat (g), LDL-C (mg/dL), and adiponectin (ng/mL) levels were significantly different between the groups. The SKO-001 group showed a decrease in body fat mass and LDL-C levels and an increase in adiponectin levels. Lean body mass, visceral fat area, body weight, BMI, waist circumference, hip circumference, waist/hip circumference ratio, TG, HDL-C, and insulin levels did not show significant differences between the groups before and after SKO-001 and placebo treatment.

These results are consistent with research showing an increase in serum adiponectin levels, a decrease in LDL-C levels, and a decrease in body fat mass in rats. Some *L. plantarum* strains have been shown to decrease the mRNA levels of adipogenic genes, including SREBP-1c, PPAR-γ, and C/EBPα [13]. Similarly, SKO-001 reduced lipid accumulation in the liver, lowered mRNA levels of lipogenic genes, and reduced α-smooth muscle actin and collagen type 1 alpha 1 levels [13].

Detailed DEXA analysis (arms, legs, trunk, male type [android], and female type [gynecoid]) was performed on the trunk and male type (android), which are expected to be related to abdominal obesity when the calorie intake was 1500 kcal. The analysis was conducted on participants with excess weight and those with physical activity of 3000 METs/w.

The Korean Society of Obesity Guidelines 2022, 8th Edition Summary, recommends a calorie intake of 1200–1500 kcal for low-calorie meals for those weighing below 113 kg and 1500–1800 kcal for those weighing over 113 kg as a dietary prescription for weight loss. Based on this evidence, we performed a stratification analysis for those who exceeded the 1500 kcal threshold [18]. The reason for conducting additional analyses in participants with a physical activity level of <3000 METs/w was based on the definition of regular physical activity practitioners; a cumulative physical activity level of 600 METs/w was defined as regular physical activity, and 3000 METs/w as vigorous and sufficient physical activity, according to the Global Physical Activity Questionnaire assessment used in this human application study in the 2020 Guidelines for Community Integrated Health Promotion (Physical Activity) [19].

In the detailed DEXA analysis, significant differences were confirmed in trunk body fat percentage, male pattern body fat percentage, and trunk body fat mass in the group comparison between the placebo and SKO-001 groups, a decrease was confirmed in the SKO-001 group, and a statistically significant difference was confirmed in the body fat percentage and body fat mass in the stratification analysis of calorie intake of >1500 kcal and physical activity of <3000 METs/w.

In clinical studies, *L. plantarum* has been shown to alter cholesterol levels in individuals with normal weight or borderline cholesterol levels [20]. In a non-alcoholic fatty liver disease mouse model fed a high-fat, high-fructose diet for 10 weeks, supplementation with *L. plantarum* prevented body weight gain, improved glucose and lipid homeostasis, reduced white fat inflammation, and reduced the progression of non-alcoholic fatty liver disease. This is believed to be the result of *Lactiplantibacillus* affecting the composition and function of intestinal microorganisms that affect white adipose tissue [21].

Adiponectin and leptin are specific to adipose tissue and are linked to obesity and insulin resistance [22]. These adipocytokines are released from adipose tissues [23,24]. Interestingly, while adiponectin levels in the bloodstream decrease as body fat increases, in contrast to the anti-obesity effect of leptin, serum leptin levels are positively correlated with body fat content. This discrepancy indicates that reduced adiponectin and elevated leptin levels are closely linked to metabolic risk factors [25].

The trial results indicated increased adiponectin levels in the SKO-001 group and decreased levels in the placebo group after SKO-001 consumption. Despite the increasing global prevalence of obesity, there is a lack of effective treatment strategies. Although various medications are presently used to address severe obesity and associated metabolic conditions [26], efforts are needed to reduce the adverse effects of these drugs.

To address the side effects of existing drugs, the focus of research on obesity treatment and body fat reduction has shifted to natural products for overcoming the side effects of existing drugs. Some probiotics have shown beneficial effects on weight loss in rodents and may be used for body fat reduction and obesity treatment in humans [27,28]. *Lactobacillus* and *Bifidobacterium* are the most well-studied probiotic strains for body fat loss and the management of obesity-related diseases [29]. Among probiotics, *L. plantarum,* a gram-positive lactic acid bacterium, stands out for its ability to produce lactic acid through the fermentation of various sugars, such as arabinose, glucose, fructose, galactose, maltose, sucrose, and dextran. Widely acknowledged to be safe, *L. plantarum* demonstrates robust probiotic attributes, leading to the commercialisation of several strains as supplements and functional foods aimed at promoting health. *L. plantarum* is widely distributed in nature and found in several habitats, including dairy products, fermented vegetables such as pickles and kimchi, fruits, and the gastrointestinal tract of mammals. In particular, during the late stage of kimchi fermentation, when kimchi is highly fermented and tastes sour, these bacteria become dominant. They suppress the growth of other species and are known to have excellent acid and bile resistance [13].

Enhancements in obesity-related comorbidities, such as weight reduction following various gastric surgeries, may be attributed to changes in the microbiome or metabolic shifts within the gut microbiota caused by surgical interventions. Food absorption in the gut is linked to metabolic disorders such as obesity, insulin resistance, and diabetes. However, the underlying molecular mechanisms remain unclear [4,5]. Disruptions in the gut microbiota provoke a gut immune response and lead to the reorganisation of gut homeostasis. These interactions not only induce dynamic changes in gut flora but also maintain the integrity of the gut barrier. Moreover, the gut microbiota participates in a systemic multiorgan communication network within the gut–brain and gut–liver axes. The absorption of high-fat diets in the gut affects the host’s feeding preferences and systemic metabolism. Interventions targeting the gut microbiota can potentially mitigate reduced glucose tolerance and insulin sensitivity associated with both central and peripheral metabolic disorders [6].

In the present study, the enrolment of participants commenced in November and concluded around March (Appendix A). The majority of the participants were enrolled between November and February, which corresponds to the autumn and winter seasons in Korea. In general, there are many factors that can cause weight gain in winter. These include reduced exercise, physiological responses to cold, and opportunities to consume high-calorie foods, for example, during winter holidays [30,31]. Research examining seasonal variations in food intake and metabolism has revealed that energy intake peaks during winter, declines notably with rising temperatures, and diminishes by approximately 25% during summer. Resting metabolic rates were the highest during winter but decreased by approximately 20% in summer. Physical activity levels tend to decrease in winter, increase notably in spring, and decline again in summer. Body weight tends to increase during winter as fat accumulates in the trunk and arms, whereas it decreases in summer as fat levels decrease. Despite the rise in the resting metabolic rate during winter, weight gain still occurs due to elevated energy intake and reduced physical activity levels [32]. Given these previous findings, it is possible that both the SKO-001 and placebo groups gained body fat in the winter months, but the interventional formulation may have caused the SKO-001 group to gain less body fat than the placebo group, and the SKO-001 group tended to lose this fat more commonly.

Several studies have reported that some probiotic strains can reduce body weight and fat mass [33], and several clinical trials have shown that probiotic intake is associated with a reduction in body weight and fat mass [34,35,36]. Although these studies have demonstrated the potential effects of certain probiotic strains on body fat reduction, more clinical studies are needed to fully understand the mechanisms underlying these effects and to identify the most effective strains, doses, and supplementation periods.

A previous experimental study on SKO-001 analysed the expression of genes involved in adipogenesis in adipose tissue to further elucidate the mechanism of action of SKO-001 [13]. The results showed that the levels of *SREBP-1c*, which increases adipogenesis [37], were significantly reduced after SKO-001 treatment in a dose-dependent manner. PPARγ is a key transcription factor that regulates glucose and lipid metabolism [38,39]. SKO-001 treatment reversed the high-fat diet (HFD)-induced increase in PPARγ expression. We also found that the mRNA expression of *C/EBPα*, a key regulator of adipogenesis and lipid accumulation [40], was completely inhibited by SKO-001, reducing lipid accumulation. C/EBPα binds to its promoter to regulate leptin expression, which plays an important role in body weight homeostasis [41]. In previous experimental studies, a consistent decrease in leptin levels was observed after SKO-001 treatment, a trend replicated in the present study. A similar inhibitory effect of SKO-001 on the differentiation of 3T3-L1 preadipocytes was observed, in parallel with a decrease in the expression of adipogenic genes, which may be the mechanism by which SKO-001 reduces body fat. Increased fat accumulation in obesity is associated with dyslipidaemia, which collectively indicates increased triglyceride, LDL-C, and TC levels and decreased HDL-C levels, increasing the risk of coronary artery disease [42].

The results of this study showed that SKO-001 improved serum lipid levels by reducing LDL-C and TC levels, with trends similar to those found in previous experimental studies. While individual responses to probiotics may vary, and factors such as diet, lifestyle, and gut microbiome composition may influence efficacy, this study provides clinical confirmation of SKO-001’s effectiveness in reducing body fat, similar to previous experimental findings. The findings of this double-blind, placebo-controlled study are significant because they provide clinical evidence that natural compounds can influence the gut environment and have a positive impact on weight loss. Furthermore, these findings may be useful in providing clinical evidence on the effects of natural ingredients on human gut microbiota metabolism.

One limitation of this study is its single-centre design, lacking the advantages of a crossover study. Future investigations conducted across multiple centres and using crossover designs could yield more definitive and reliable evidence. In addition, microbiome analysis was not attempted in the trial due to budgetary issues and subject discomfort. A limitation of this study is that faecal testing cannot provide accurate information regarding changes in the gut microbiome. This limitation hinders our ability to discern the mechanisms by which changes in the gut microbiota contribute to fat loss. It is imperative to conduct studies to validate changes in intestinal microbes following SKO-001 consumption, as well as mechanistic studies elucidating the effects of SKO-001 on reducing body fat. This trial was conducted over a relatively short period. If a larger study with a longer duration of ≥3 months is conducted in the future, we believe that the seasonal and age-specific endpoints will show more significant changes.

## 5. Conclusions

This 12-week double-blind, placebo-controlled clinical trial on the effects of SKO-001 consumption in adult men and women aged >19 years and <65 years indicated a significant difference in the change in body fat percentage between the SKO-001 and placebo groups. In particular, body fat percentage increased in the placebo group, whereas it showed a tendency to decrease in the intervention group. Among the secondary endpoints, differences in changes in body fat mass, LDL-C, and adiponectin between the groups were statistically significant. In the SKO-001 group, body fat mass and LDL-C levels decreased, and adiponectin levels increased. The reduction in body fat, body fat ratio, body fat mass, and LDL-C levels decreased in the SKO-001 group compared with that in the placebo group, but adiponectin levels increased, confirming the beneficial effect of SKO-001 on body fat reduction. SKO-001 did not cause any severe adverse reactions. In conclusion, SKO-001 is safe and effective in reducing body fat and has the potential for further testing in humans.

## Figures and Tables

**Figure 1 nutrients-16-01137-f001:**
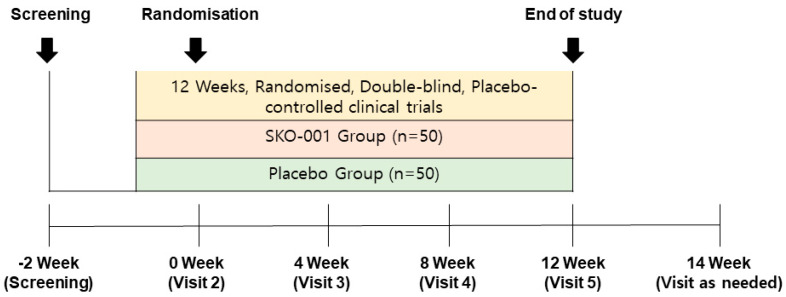
Clinical trial progression.

**Figure 2 nutrients-16-01137-f002:**
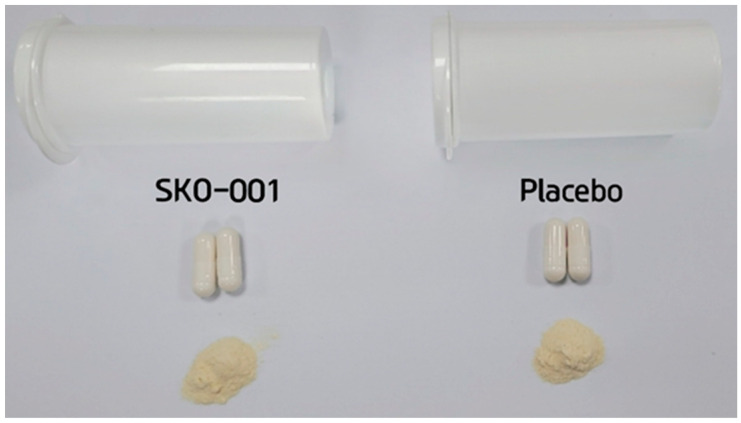
Interventional and placebo formulations consumed by the participants.

**Figure 3 nutrients-16-01137-f003:**
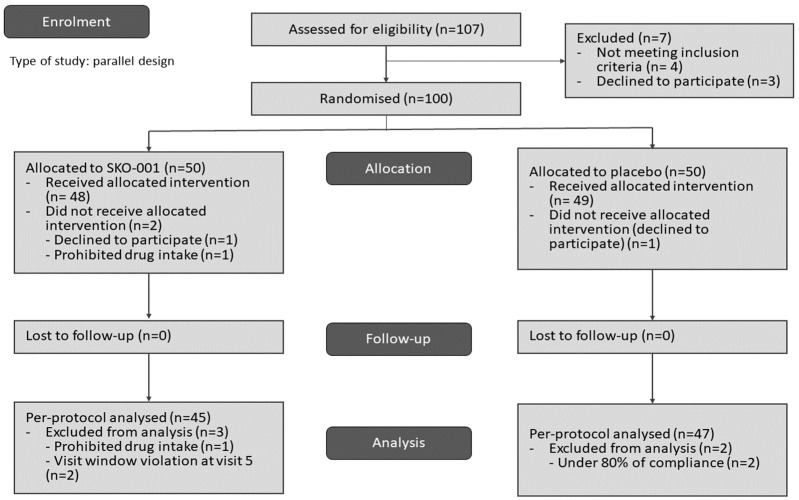
Trial flowchart. Abbreviations: SKO-001—*Lactiplantibacillus plantarum* SKO-001-based interventional formulation.

**Figure 4 nutrients-16-01137-f004:**
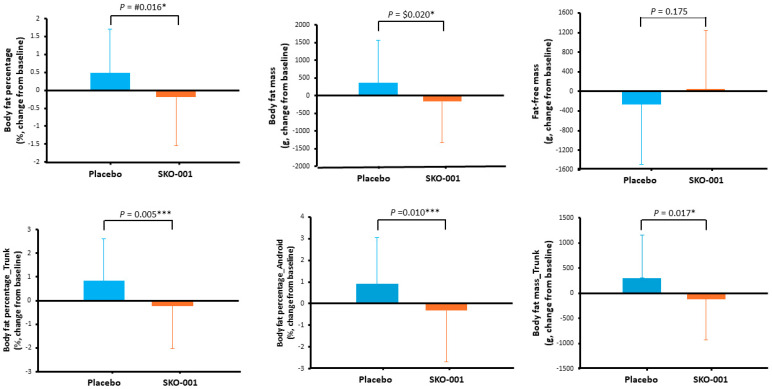
DEXA results among endpoints at 12 weeks from baseline according to group. DEXA, Dual-energy X-ray absorptiometry; SKO-001: *Lactiplantibacillus plantarum* SKO-001-based interventional formulation. ** p* < 0.05, *** *p* < 0.001. #: Independent *t*-test $: Wilcoxon rank sum test.

**Table 1 nutrients-16-01137-t001:** Participant characteristics by group.

		SKO-001 Group (N = 50)	Placebo Group (N = 50)	*p*-Value
Sex	Men	9	6	0.575 ^1^
	Women	41	44	
Age, years		48.90 ± 10.60	47.14 ± 9.67	0.248 ^2^
Height, cm		161.13 ± 7.67	161.19 ± 7.26	0.971 ^2^
Weight, kg		71.19 ± 8.25	71.64 ± 7.67	0.781 ^3^
Body mass index, kg/m^2^		27.34 ± 1.49	27.51 ± 1.40	0.539 ^2^

^1^ *p*-value for the chi-square test, ^2^ *p*-value for the Wilcoxon rank sum test, and ^3^ *p*-value for the two-sample *t*-test. Data are presented as N or mean ± SD. Abbreviations: SKO-001—*Lactiplantibacillus plantarum* SKO-001-based interventional formulation, SD—standard deviation.

**Table 2 nutrients-16-01137-t002:** Lipid profile and hormonal outcomes among endpoints at 12 weeks from baseline according to group.

Parameter	Treatment	Baseline	12 Weeks	*p*-Value	*p*-Value
Total cholesterol (mg/dL)	SKO-001	216.44 ± 36.95	204.67 ± 35.78	0.003 **	0.158
	Placebo	208.17 ± 35.47	203.62 ± 34.22	0.194	
	*p*-value	0.277	0.886		
TG (mg/dL)	SKO-001	115.42 ± 40.43	126.33 ± 62.87	0.221	0.055
	Placebo	125.13 ± 77.42	111.19 ± 62.13	0.079	
	*p*-value	0.77	0.248		
HDL-C (mg/dL)	SKO-001	60.33 ± 11.89	60.62 ± 12.17	0.953	0.191
	Placebo	58.23 ± 12.43	60.06 ± 13.47	0.079	
	*p*-value	0.239	0.705		
LDL-C (mg/dL)	SKO-001	136.91 ± 30.78	130.60 ± 32.12	0.039 *	0.025 *
	Placebo	131.38 ± 32.98	134.62 ± 33.11	0.282	
	*p*-value	0.408	0.556		
Adiponectin (ng/mL)	SKO-001	10,197.11 ± 5733.93	10,651.47 ± 6458.85	0.548	0.023 *
	Placebo	10,224.15 ± 5248.66	9595.95 ± 5120.76	0.015 *	
	*p*-value	0.828	0.581		
Leptin (ng/mL)	SKO-001	26.68 ± 13.76	22.65 ± 12.84	0.014 *	0.542
	Placebo	26.61 ± 13.20	23.03 ± 12.88	0.093	
	*p*-value	0.95	0.96		
Insulin (mIU/L)	SKO-001	7.84 ± 6.13	6.34 ± 5.55	0.084	0.687
	Placebo	5.83 ± 2.80	6.17 ± 10.01	0.063	
	*p*-value	0.337	0.617		

SKO-001: *Lactiplantibacillus plantarum* SKO-001-based interventional formulation; LDL-C, low-density lipoprotein-cholesterol; HDL-C, high-density lipoprotein-cholesterol; TG, triglyceride. ** p* < 0.05, ** *p* < 0.01.

**Table 3 nutrients-16-01137-t003:** CT results among endpoints at 12 weeks from baseline according to group.

Parameter	Treatment	Baseline	12 Weeks	*p*-Value	*p*-Value
Visceral fat (cm^2^)	SKO-001	125.91 ± 34.33	126.57 ± 48.96	0.247	0.081
	Placebo	122.98 ± 39.29	126.45 ± 41.33	0.196	
	*p*-value	0.703	0.685		
Subcutaneous fat (cm^2^)	SKO-001	223.23 ± 64.91	208.09 ± 62.63	0.008 **	0.995
	Placebo	231.83 ± 48.19	216.64 ± 55.87	0.002 **	
	*p*-value	0.196	0.327		
Abdominal fat (cm^2^)	SKO-001	349.14 ± 75.90	334.66 ± 85.46	0.059	0.762
	Placebo	354.81 ± 56.64	343.09 ± 67.57	0.028	
	*p*-value	0.687	0.602		
VSR	SKO	0.59 ± 0.22	0.64 ± 0.27	0.073	0.41
	Placebo	0.55 ± 0.25	0.62 ± 0.30	0.001 **	
	*p*-value	0.309	0.652		

CT, Computed tomography; SKO-001, *Lactiplantibacillus plantarum* SKO-001-based interventional formulation; VSR, visceral fat/subcutaneous fat area ratio. ** *p* < 0.01.

**Table 4 nutrients-16-01137-t004:** Symptom severity of adverse events and their relevance to the interventional formulation (safety set).

		SKO-001	Placebo	Total	*p*-Value ^&^
Severity	Mild	46	44	90	0.495
	Moderate	0	1	1	
	Severe	0	0	0	
	SAE (non-fatal)	0	1	1	
	SAE (fatal)	0	0	0	
Results	Complete healing (no aftereffects)	45	40	85	0.111
	Healing (with sequelae)	0	0	0	
	In progress	1	6	7	
	Permanent damage	0	0	0	
	Death	0	0	0	
Relevance	Definitely related	0	0	0	1.000
	Probably related	0	0	0	
	Possibly related	0	0	0	
	Possibly not related	1	0	1	
	Definitely not related	45	46	91	
	UK, unassessable	0	0	0	
Interventional formulation-related actions	None	6	10	16	0.283
	Dose change/pause	0	0	0	
	Interruption of intake	0	0	0	
	Combination drug intake	40	35	75	
	Non-drug treatment	0	1	1	
	Increased length of hospitalisation	0	0	0	

Data are presented as N. SKO-001: *Lactiplantibacillus plantarum* SKO-001-based interventional formulation. ^&^ Fisher’s exact test.

## Data Availability

The data presented in this study are included in the article/Appendix A. Further inquiries can be directed to the corresponding author.

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
