# Peer review of "A 12-Week, Single-Centre, Randomised, Double-Blind, Placebo-Controlled, Parallel-Design Clinical Trial for the Evaluation of the Efficacy and Safety of Lactiplantibacillus plantarum SKO-001 in Reducing Body Fat"

_nutrients, 2024, doi:10.3390/nu16081137_

Round 1

Reviewer 1 Report

Comments and Suggestions for Authors

In the present study, the authors have evaluated the effect of L. plantarum SKO-001 on reducing body fat through RCT. The strain has been predicted to improve obesity-related factors in vitro and in vivo analyses. Although the present RCT study seems to be appropriately designed and conducted, however, there are some concerns in the manuscript for acceptance at present form as follows:

1.

  The discussion section seems to consist of consideration in each result, thus the authors should write those more comprehensively with previous reports in the field of health-promoting effects of probiotic strains and in the viewpoint of mechanism of probiotics and its products (called as postbiotics).

2.

  Why the authors use different statistical methods in test and placebo groups on intra-group analyses, e.g., Body fat mass in Table 3.  It seems incomprehensible.  Further, the authors mentioned that they used ANCOVA on inter-group analyses using the baseline as covariate, however some items were analyzed by other method, e.g., Body fat percentage in Table 3.

3.

  In the result section, there are many descriptions on the comparison in other measured items, therefore, some important results should be inserted in the main text (not as Supplementary Table).

4.

  Regarding adverse-effect monitoring, the authors should summarize all of the observed undesirable adverse events by each grade (with results of Fisher’s exact test) through the clinical trial period according to the latest version of CTCAE, and the result (Table) should be placed in the main text.

5.

  e.g., line 492–, not all L. plantarum strains show those properties, but some strains so.

6.

  Did the authors confirm the differences in intake of calories between two groups?

Reviewer 2 Report

Comments and Suggestions for Authors

Well-written article. Some minor 

1. No need to add the inclusion and exclusion criteria in a table. Instead, add it as a section and write it as text.

2. Lines 121-123, please rewrite this in a clearer way. It is confusing with respect to the component of each supplement. 

3. Figure 1 is okay, but could be replaced with a more useful figure for the reader. You could represent your study design as a figure showing the timeline and visits involved in the study and the tests done. 

4. Line 144 - check assessed by ?

5. I would replace Table 3 with a figure so it becomes easier for the reader to grasp the key findings. You are already mentioning the values in the text, so a graph will be more useful than a figure. The current table can be moved to the supplementary file. 

Reviewer 3 Report

Comments and Suggestions for Authors

Yours is a potentially important study that deserves to be shared. The critical question is why, for most of the indices, the probiotic group did not show significant change whereas the placebo group showed an increase in body fat % as well as body fat mass. Overall, the difference in the change between the two groups was significant. You mention that most of the individuals started the study during a time of year when some seasonal weight gain is common. Is there a sufficient distribution of patients to examine that? In other words, is there a sufficient number of participants in each group to compare those who started in the spring v. those who started in the fall? At the very least you would need to ensure that the seasonal distribution of probiotic and placebo treatment initiations was not significantly different.

There is a great deal of data in the Results section. I would suggest that the authors consider some bar graphs showing the change in each group and significance. A visual presentation would be easier for the readers to appreciate.

Other issues: Previous studies 58 have reported that the probiotic Lactiplantibacillus plantarum can ameliorate obesity and 59 improve weight or body fat-related indicators, possibly via the regulation of lipid metabolism [12-13]. Please indicate that these were studies in mice. You might want to comment on comparable studies in man, whether for weight gain or other targets.

Advertisement: please state briefly how the study was described in the advertisement and provide a copy of the advertisement, translated into English in the Supplementary Information. Were participants provided with a financial incentive?

Exclusion criteria: Was recent history of antibiotic use considered? What about a history of GI disturbances such as irritable bowel or inflammatory bowel disease?

Were the participants asked to ‘guess’ as to whether they were doses with active probiotic or placebo?

Can you provide the educational level or any information on the socioeconomic status of the participants?

Statistics: How many of the findings survive a correction for multiple-comparisons?

Given the relatively large number of adverse reactions, please add more information as to their nature in Table S9. In 75 cases drugs were administered. Was the use of drugs comparable in the two groups?

You state “There were no significant between-group differences in dietary intake before (0

470 weeks) and after (12 weeks) SKO-001 consumption”. What about in the placebo group?

“faecal testing could not provide accurate information about changes in the gut microbiome.” Does this mean that you attempted to assay the fecal microbiome and did not find any changes or that you did not attempt to assay the faecal microbiome?

Are the changes that you detected clinically significant? Even if they are not, it would be worth underscoring that over a longer period of time, they could be.

It would also be important to provide some additional information about methods.

Round 2

Reviewer 1 Report

Comments and Suggestions for Authors

The revised version of the manuscript seems to be well refined according to the reviewers' comments.

Just one point, if different statistical analyses were used and the results were summarized in same table, each method should be described in the table as previous version of the manuscript.

Author Response

Thank you so much for your feedback.

The authors have made the following changes based on your feedback. 

As per your revision instructions, I have marked the different statistical methods used in the main body of the paper(Figure 4) and in the Supplement(Table S12, S15) in red. 

Reviewer 3 Report

Comments and Suggestions for Authors

Lines 53-55: Specify species

Lines 83-84: Specify that the solicitation indicated that the goal of the study was to test a dietary supplement for body fat reduction.

Somewhere you should comment that the sample was likely composed of those motivated to lose body fat.

Lines 134-135: Indicate the duration of the antibiotic-restricted period prior to enrollment.

Author Response

Thank you so much for your feedback.

The authors have made the following changes based on your feedback. 

Lines 53-55: Specify species

⇒We've added the following to the body 

"Previous studies have reported that the probiotic Lactiplantibacillus plantarum, SKO-001, K50, KY1032, LMT1-48 can reduce obesity and improve weight- or body fat-related indicators, possibly via the regulation of lipid metabolism"

Lines 83-84: Specify that the solicitation indicated that the goal of the study was to test a dietary supplement for body fat reduction.

Somewhere you should comment that the sample was likely composed of those motivated to lose body fat.

⇒We've added the following to the body 

" This protocol (version 1.2, publication date, January 1, 2023) was approved by the Institutional Review Board of Seymour University Oriental Medicine Hospital (approval number SMJOH-2022-08) and registered with the Korea Clinical Research Information Center (KCT0008871) to test the effectiveness of dietary supplements in reducing body fat."

Lines 134-135: Indicate the duration of the antibiotic-restricted period prior to enrollment.

⇒ We've added the following to the body 

"We did not enroll those who took antibiotics within 4 weeks of starting the study, and those who took antibiotics after 4 weeks were enrolled in the study"